# Effect of *Artemisia absinthium* and *Malva sylvestris* on Antioxidant Parameters and Abomasal Histopathology in Lambs Experimentally Infected with *Haemonchus contortus*

**DOI:** 10.3390/ani11020462

**Published:** 2021-02-09

**Authors:** Dominika Mravčáková, Małgorzata Sobczak-Filipiak, Zora Váradyová, Katarína Kucková, Klaudia Čobanová, Peter Maršík, Jan Tauchen, Jaroslav Vadlejch, Marcin Mickiewicz, Jaroslaw Kaba, Marián Várady

**Affiliations:** 1Institute of Animal Physiology, Centre of Biosciences, Slovak Academy of Sciences, 040 01 Košice, Slovakia; varadyz@saske.sk (Z.V.); kuckova@saske.sk (K.K.); boldik@saske.sk (K.Č.); 2Department of Pathology and Veterinary Diagnostics, Institute of Veterinary Medicine, Warsaw University of Life Sciences, 02-776 Warsaw, Poland; malgorzata_sobczak_filipiak@sggw.edu.pl; 3Institute of Experimental Botany, Czech Academy of Sciences, Prague, 165 02 Lysolaje, Czech Republic; marsik@af.czu.cz; 4Department of Food Science, Faculty of Agrobiology, Food and Natural Resources, Czech University of Life Sciences, 165 00 Prague-Suchdol, Czech Republic; tauchen@af.czu.cz; 5Department of Zoology and Fisheries, Faculty of Agrobiology, Food and Natural Resources, Czech University of Life Sciences, 165 00 Prague-Suchdol, Czech Republic; vadlejch@af.czu.cz; 6Division of Veterinary Epidemiology and Economics, Institute of Veterinary Medicine, Warsaw University of Life Sciences, 02-776 Warsaw, Poland; marcin_mickiewicz@sggw.edu.pl (M.M.); jaroslaw_kaba@sggw.edu.pl (J.K.); 7Institute of Parasitology, Slovak Academy of Sciences, 040 01 Košice, Slovakia

**Keywords:** abomasum, antioxidant parameters, *Artemisia absinthium*, gastrointestinal nematode parasite, *Haemonchus contortus*, histopathological changes, local immune response, *Malva sylvestris*

## Abstract

**Simple Summary:**

*Haemonchus contortus* is a blood-feeding gastrointestinal nematode (GIN) that parasitizes in the abomasum of small ruminants. Infections caused by this parasitic nematode possess a serious threat to livestock production worldwide, and with the expansion of anthelmintic resistance, there is an urgent need for more sustainable alternative controls of GIN. It is known that nutritional supplementation with medicinal plants could slow the dynamics of infection and increase the resistance of lambs to parasitic infection. The influence of medicinal plants used for control of haemonchosis on the local immune response of abomasal mucosae in GIN-infected sheep, however, has not been reported. This study aimed to evaluate the effect of diets containing wormwood (*Artemisia absinthium*)*,* mallow (*Malva sylvestris*), or their mix, on antioxidant parameters and local immune responses in the abomasum of lambs infected with *H. contortus.* Treatment with the medicinal plants affected antioxidant parameters by reducing oxidative stress in the abomasal mucosa and helped trigger local immune responses. Wormwood (*A. absinthium*) and mallow (*M. sylvestris*) applied as dietary supplements may increase the resistance of lambs to GIN infections.

**Abstract:**

This study evaluated the effect of *Artemisia absinthium* and *Malva sylvestris* on antioxidant response and histopathological changes in the abomasa of the *Haemonchus contortus* infected lambs. Twenty-four lambs were divided into four groups: unsupplemented lambs (UNS), lambs supplemented with *A. absinthium* (ART), lambs supplemented with *M. sylvestris* (MAL), and lambs supplemented with both plants (ARTMAL). Lambs were infected orally with approximately 5000 third-stage (L3) larvae of *H. contortus*. The experiment was conducted for 75 d (days), all animals were then slaughtered, and the abomasal tissues were examined for antioxidant parameters and histopathology. The concentration of malondialdehyde in the abomasal mucosa was lower in ARTMAL (*p* < 0.05), and the total antioxidant capacity was higher in MAL (*p* < 0.05), than in UNS. Increased mucus production was observed in the ARTMAL. The number of mast cells in UNS and ART was significantly higher than the number in MAL (*p* < 0.01 and *p* < 0.05). Plasma cell numbers were higher in ARTMAL than the number in MAL (*p* < 0.05). Abomasal tissue regenerated more frequently in ARTMAL. These results represent the first report of the impact of *A. absinthium* and *M. sylvestris* on antioxidant parameters and local immune responses of abomasal mucosa of lambs infected with a GIN parasite.

## 1. Introduction

Gastrointestinal nematodes (GINs) represent a major threat to the small-ruminant industry worldwide. *Haemonchus contortus* is one of the most pathogenic GINs, mainly because of its blood-feeding behavior. Infections with this parasite are associated with negative impacts on animal health, welfare, and production, which can lead to increased mortality, predominantly in young animals [1]. The control of GIN infections is usually limited to the frequent and repeated use of anthelmintic drugs. Their excessive use, however, has led to the development of anthelmintic resistance in GIN populations [2] and has increased the level of chemical residues in animal products [3]. The expansion of anthelmintic resistance and consumer demand for chemical free animal products has created an urgent need for a more sustainable alternative control of GINs [4,5], such as nutritional supplementation with medicinal plants [6,7,8].

Medicinal plants contain various bioactive compounds, such as alkaloids, flavonoids, terpenoids, lactones, and glycosides, that have different modes of therapeutic action, e.g., immunomodulatory, anti-inflammatory, antioxidant, or anthelmintic [9,10,11,12]. Some secondary metabolites of plants have direct anthelmintic activity against GINs [13,14,15]. Several studies are currently dealing with bioactive compounds with the anthelmintic activity of organic fractions obtained from plant extracts used as natural anthelmintics against *H. contortus* eggs and infective larvae [16,17,18]. However, the use of whole plants as dietary supplements in feeds can lead to different results. Our recent study [19] showed that the *A. absinthium* and *M. sylvestris* possess a strong anthelmintic effect in vitro versus a slight effect in vivo. On the other hand, the variety and synergy of bioactive compounds of several medicinal plants and their different combinations together described in our previous studies can contribute to a certain pharmacological efficacy [20,21,22]. These studies concluded that supplementation with a mixture of dried medicinal plants in the diet of lambs could slow the dynamics of infection and increase the resistance of lambs to parasitic infection.

The supplementation of diets with plant nutraceuticals containing bioactive substances has received increasing attention for manipulating host nutrition and indirectly improving animal resistance to GIN infections [23,24,25]. Nutraceuticals can provide enough nutrients essential for tissue maintenance and blood homeostasis, but also for host immune responses against parasites [7], which can affect the overall health of animals. Immunity to nematodes is associated with innate and acquired responses, but the local cellular immune responses in host abomasa are considered the most important defenses against haemonchosis. Mucosal mast-cell hyperplasia, the formation of globular leukocytes, eosinophilia (an increase in eosinophils in blood and tissue), increased mucus production in the abomasum, and the production of specific antibodies typically emerge during infection [26,27,28]. The presence of nematodes in the abomasum can also induce the production of reactive oxygen species by immune cells, which can damage the parasites and cause oxidative stress in the host [29]. Plant secondary metabolites may improve the health of animals through their antioxidant and immune-enhancing properties [30,31]. The mechanisms of sheep resistance to GINs via plant secondary metabolites, however, are not fully understood.

Many studies have investigated alterations in sheep abomasa in association with *H. contortus* infection [32,33,34]. To our knowledge, the influence of medicinal plants on antioxidant parameters and histopathological changes in sheep abomasa have not been reported. We hypothesized that dietary supplementation with medicinal plants would influence the local immune response in the abomasum and positively affect the antioxidant status of infected animals. This study aimed to evaluate the effect of diets containing wormwood (*A. absinthium*) and mallow *(M. sylvestris*) (1) on antioxidant parameters in sera and abomasal mucosa, and (2) on histopathological changes in the abomasal tissues of lambs experimentally infected with *H. contortus*.

## 2. Material and Methods

### 2.1. Ethics Statement

The experimental protocol was approved by the Ethical Committee of the Institute of Parasitology of the Slovak Academy of Sciences, following national legislation in Slovakia (G.R. 377/2012; Law 39/2007) for the care and use of research animals. Permission to collect samples and to carry out the experiments was granted by the participating sheep farmers.

### 2.2. Animals and Experimental Design

The experiment was part of a larger study that evaluated the effect of dry *Artemisia absinthium* and *Malva sylvestris* plants on the parasitic gastrointestinal nematode *H. contortus* in lambs, which has previously been described in more detail [19]. Briefly, twenty-four 3–4 month-old female lambs (improved Valachian) with initial body weights of 18.67 ± 0.55 kg were housed in common stalls on a commercial sheep farm (Hodkovce, Slovakia) for 14 days (d) for acclimatization to the feeding and maintained in their production system during the experiment with free access to water. After a period of adaptation, all parasite-free lambs were infected orally with approximately 5000 third-stage (L3) larvae of a strain of *H. contortus* (MHCo1) susceptible to anthelmintics [21]. Each animal was fed oats (500 g dry matter (DM)/d) and meadow hay (*ad libitum*). Infected lambs were randomly divided into four groups of six animals each (six lambs/group, one stall/group) based on their live weights: unsupplemented lambs (UNS), lambs supplemented with dried *A. absinthium* stems (ART, 1 g DM/d/lamb), lambs supplemented with dried *M. sylvestris* (MAL, 15 g DM/d/lamb) flowers, and lambs supplemented with a mixture of dried *A. absinthium* stems and *M. sylvestris* flowers (ARTMAL, 16 g DM/d/lamb). The number of animals used in the experiment was assigned following VICH GL13 guidelines proposed by the European Medicines Agency. The dried plant components of *A. absinthium* and *M. sylvestris* both from commercial sources (AGROKARPATY, Plavnica, Slovakia) were mixed daily with the oats during the experimental period (75 d). Qualitative phytochemical screening of active compounds in aqueous plant extracts identified mainly tannins and saponins (*A. absinthium*) and tannins, flavonoids, glycosides, alkaloids, and terpenoids (*M. sylvestris*) [19]. The phytochemical substances of *A. absinthium* contained 6.48 g/kg DM phenolic acids and 0.35 g/kg DM flavonoids, *M. sylvestris* contained 0.65 g/kg DM phenolic acids and 6.48 g/kg DM flavonoids [19]. Blood samples for sera were collected from each animal at day 75 from the jugular vein by using a 21-gauge needle and syringe into 10 mL serum separator tubes (Sarstedt AG & Co, Nümbrecht, Germany) and centrifuged at 1200× *g* for 10 min at room temperature. The sera were stored at −80 °C until analysis. All animals were humanely killed on 75 d after infection, (abattoir of the Centre of Biosciences of SAS, Institute of Animal Physiology, Košice, Slovakia, No. SK U 06018), and helminthological dissections were performed [21]. The carcasses were sent to the Department of Pathological Anatomy and Pathological Physiology, University of Veterinary Medicine and Pharmacy in Košice in the Slovakia. Moreover, samples of their abomasa were collected for the evaluation of antioxidant parameters and histopathological examination.

### 2.3. Plant Methanolic Extracts

We separately dissolved 0.248 g of *A. absinthium* and 0.262 g of *M. sylvestris* in 1 mL of methanol (MeOH) to the final concentrations of 250 mg/mL and then centrifuged the extracts to separate the insoluble fractions. The supernatants were used for further fractionation. Some fractions were prepared using a Dionex Ultimate 3000 semi-preparative high-performance liquid chromatograph (HPLC) with a binary HPLC pump, autosampler, photodiode array (PDA) detector, fraction collector (AFC-3000), column-temperature controller, and mobile-phase degasser (Thermo Scientific, Sunnyvale, CA, USA). The fractions were separated on a C18 column (15 µm, 280 × 10 mm (Phenomenex, Torrance, CA, USA), tempered at 26 °C with mobile phases of water (A) and MeOH (B) by gradient elution. The initial gradient was 5% phase B (from 0 to 5 min), increasing to 100% B in 40 min, which was maintained for 5 min and was followed by 10 min of equilibration at an initial level of 5% B from 47 to 57 min. The flow of the mobile phase was 9 mL/min. An injection volume of the methanolic extract was 100 µL. The separation was measured as an absorption intensity of spectra monitored within a range of 190–400 nm. The analytical outputs were acquired and processed using Chromeleon 7.2.8 (Thermo Scientific Dionex, Sunnyvale, CA, USA). Some fractions were evaporated to dryness using a vacuum rotary evaporator (Büchi rotavapor 100, Flawil, Switzerland) at 40 °C, redissolved in 1 mL of 100% MeOH, moved to 2-mL vials, and again evaporated to dryness in a stream of nitrogen at 35 °C. The evaporating flask was then washed three times with 1.5 mL of MeOH, and combined volumes were added to the previous fraction and evaporated using nitrogen. Dry samples were stored at −18 °C.

### 2.4. Antioxidant Parameters

The folds of the abomasa were flushed with saline, and the mucosae were subsequently sampled by scraping the inside surfaces with a glass slide and stored at −80 °C until analysis. Tissue processing and antioxidant assays have been previously described in more detail [35]. The activity of glutathione peroxidase (GPx) in the serum and abomasal mucosa was determined spectrophotometrically as described by [36]. One unit of enzymatic activity was equal to 1 µmol of NADPH oxidised per minute at 340 nm and is expressed per gram of tissue protein or milliliter of serum. The total antioxidant capacity (TAC) of the serum and abomasal mucosa was determined using a ferric reducing antioxidant power assay following the method of [37]. The final results are expressed in mmol Fe2^+^/L of serum or µmol Fe2^+^/g of tissue protein. The malondialdehyde (MDA) concentration in the serum and tissue homogenate was measured using a modified fluorometric method as described by [38]. Protein concentrations in the tissues were determined spectrophotometrically as described by [39].

### 2.5. Abomasum Histology

Each abomasum was removed after necropsy, opened along the greater curvature, and gently washed with water to recover nematodes. Two tissue samples from the fundic part of the abomasum (approximately 3 × 3 cm) were removed and routinely fixed in 10% neutral buffered formalin, embedded in paraffin (Paraplast, Sigma-Aldrich, Steinheim, Germany), and subsequently cut with a rotatory microtome into sections 4 µm thick. The paraffin slides were automatically stained with haematoxylin and eosin (Varistain Gemini Thermo Scientific, Leicestershire, UK) for histopathological examination and counting eosinophils, plasma cells, and mitotic figures (to determine the mitotic activity of the epithelial cells at the neck of glands). Mucus was stained with Periodic acid-Shiff stain, and mast cells were stained with toluidine blue. Eosinophils, plasma cells, mast cells, and mitotic figures in the epithelial cells of the glands in the abomasal mucous membrane were counted for each animal in 10 randomly selected fields at an objective magnification of 400× using an Olympus BX43 light microscope equipped with an SC30 digital camera (Olympus Optical, Tokyo, Japan). Photomicrographs were recorded and analysed using CellSens Entry 2011 (Olympus Lifescience, Tokyo, Japan).

### 2.6. Calculations and Statistical Analysis

Data for the antioxidant parameters and immune cells in the abomasal mucosa were evaluated by multiple comparisons using one-way ANOVAs. Differences between the groups in both analyses were evaluated using Bonferroni post hoc tests. Student’s *t*-tests were applied to assess the differences between the mean numbers of immune cells for the groups on day 75. The significance level was set at *p* < 0.05.

## 3. Results

### 3.1. Plant Methanolic Extracts

The methanolic extracts of wormwood, *A. absinthium*, and mallow, *M. sylvestris*, were each divided into four fractions (Table 1, see chromatograms as Appendix A). The fractions of the *A. absinthium* methanolic extract contained the highest concentrations of very polar substances, saccharides, peptides, and polar phenolic acids (46,000 µg/mL), followed by polyphenols, less polar phenolic acids, and sesquiterpenes (20,000 µg/mL). Nonpolar substances, lipids, resins, and di- and triterpenes had concentrations of 12,000 µg/mL, and low-polarity polyphenols, phenolic acids, and sesquiterpenes had concentrations of 6000 µg/mL. The fractions of the *M. sylvestris* methanolic extract contained 150,000 µg/mL saccharides and polar phenols, 42,000 µg/mL anthocyanins and other phenolic compounds, 18,000 µg/mL nonpolar substances, esters, fatty acids, and lipids, and 9000 µg/mL polyphenols, flavonoids, and phenolic acids.

### 3.2. Antioxidant Parameters

Table 2 presents the antioxidant parameters in the abomasal mucosae and sera of infected lambs on day 75. The concentration of MDA in the abomasal mucosa was significantly lower (*p* < 0.05) in ARTMAL than UNS. TAC in the abomasal mucosa was significantly higher (*p* < 0.05) in MAL compared to UNS. The serum antioxidant parameters were not affected by the treatments (*p* > 0.05).

### 3.3. Abomasum Histology

Mean worm abomasal counts were 899 ± 95, 984 ± 382, 876 ± 269 and 787 ± 335 for UNS, ART, MAL and ARTMAL, respectively [19]. Histopathological changes were predominantly in the mucosal membrane in all groups. All animals had infiltrates of inflammatory cells (mainly lymphocytes and macrophages, admixed with eosinophils, plasma cells, and mast cells), with the lowest intensity in UNS (Figure 1a) and the highest intensity in ARTMAL (Figure 1b). Inflammatory infiltrates were in all layers of the mucous membrane, i.e., in the superficial and deep zones, in all groups except UNS. The infiltrates in UNS were primarily at the bases of the glands, in the deep zone. Lymphoid nodules formed in some cases, mostly in MAL (Figure 1c). Mucus in UNS, MAL, and ART was produced predominantly in the superficial part of the mucosal membrane (Figure 1d). Mucus production (throughout the thickness of the mucosa), though, was highest in ARTMAL, which caused the distension of some glands in this experimental group (Figure 1e).

### 3.4. Immune Cells in the Abomasal Mucosa

Figure 2 shows the mean numbers of eosinophils (a), mast cells (b), plasma cells (c), and mitotic figures (d) on day 75 in the abomasal mucosae of the lambs in all experimental groups. The number of eosinophils in the abomasal mucosa did not differ significantly between the groups (Figure 2a). The number of mast cells was significantly higher in UNS and ART (*p* < 0.01 and 0.05, respectively) than MAL (Figure 2b). The mean number of plasma cells was significantly higher (*p* < 0.05) in ARTMAL than MAL (Figure 2c). The number of mitotic figures (the mitotic activity of the epithelial cells at the neck of glands; Figure 1f) was significantly higher in ARTMAL than MAL (*p* < 0.05) and UNS and ART (*p* < 0.001). Mitosis was more frequent in MAL (*p* < 0.05) than UNS and ART (Figure 2d).

## 4. Discussion

This experiment was part of a study investigating the anthelmintic activity of *A. absinthium* and *M. sylvestris* against *H. contortus* in lambs [19]. The present part of the study investigated the indirect increase in resistance of lambs infected with *H. contortus* by supplementing their diets with *A. absinthium* and *M. sylvestris*. The fractionation of the methanolic extracts indicated that *M. sylvestris* contained more saccharides, polar phenols, and a higher concentration of flavonoids than did *A. absinthium*. *M. sylvestris* flowers are rich in bioactive compounds, such as phenols, flavonoids, and carotenoids, which are characterised by their powerful antioxidant properties [40]. In contrast, *A. absinthium* stems contained proteins, sesquiterpenes, di- and triterpenes, and resins, but lower concentrations of phenolic acids and polyphenols. *A. absinthium*, however, also contains many phytochemical compounds such as lactones, terpenoids, essential oils, resins, tannins, and phenols that have antioxidant, antimicrobial, anthelmintic, and immunomodulatory activity [41]. More complex studies are probably necessary before excluding the direct anthelminthic properties of these plants in vivo [19] because the complexity of the physiology of the gastrointestinal system of ruminants makes it particularly difficult to evaluate the potential efficacy of the same plant when administered fresh after the harvest or after its drying. In addition, promising results are also shown by exploring the potential anthelmintic effect of fractions of bioactive compounds from extracts of some plants (e.g., *Acacia cochliacantha, A. farnesiana, Prosopis laevigata)* in vitro or in vivo [16,17,18].

The presence of nematodes in sheep abomasa is usually associated with the production of reactive oxygen species by immune cells, which can represent host defensive mechanisms against parasites, but can also generate oxidative stress [42]. Oxidative stress arises as an imbalance between the formation of oxidants and antioxidants in the body and can damage lipids, DNA, proteins, and carbohydrates, which eventually leads to tissue damage and inflammation [29]. Traditional medicinal plants have long been known for their antioxidant properties. Amongst all secondary metabolites in plants, phenolic compounds, which are present in both *A. absinthium* and *M. sylvestris* [19], are considered to be the strongest natural antioxidants [43]. The analysis of the antioxidant parameters indicated that dietary supplementation with two medicinal plants in ARTMAL reduced the MDA level in the abomasa of infected sheep by day 75, which could indicate the inhibition of lipid peroxidation and the reduction of oxidative stress in abomasal tissue. MDA levels tend to increase during parasitic infection, due to cellular injury in tissues and are an indicator of oxidative stress [44]. The antioxidant activity of *A. absinthium* has been previously reported [41] and has been attributed to an abundant phenolic content. Bora and Sharma [45] reported that *A. absinthium* had strong antioxidant properties and could be used to prevent oxidative stress during disease by reducing lipid peroxidation. *M. sylvestris* also has strong antioxidant properties represented by the reduction of reactive oxygen species and inhibition of lipid peroxidation [40]. MDA levels in our study were significantly lower only in ARTMAL compared to UNS, so the combination of these two herbs may be necessary to reduce oxidative stress. Interestingly, TAC in the abomasum was significantly higher only in MAL. This parameter represents all antioxidants present in tissue or plasma and provides an integrated parameter of the status of antioxidants. Our results indicated that supplementing diets with these medicinal plants could reduce oxidative stress and affect the antioxidant status of lambs infected with *H. contortus*. The parameters of serum antioxidants, however, were not affected by the treatments, so we suggest that these two medicinal plants only affect antioxidant parameters locally.

Protective immune responses against *H. contortus* are associated with the infiltration of inflammatory cells, such as lymphocytes, macrophages, eosinophils, mast cells, and plasma cells, into the abomasal mucosa [28]. Our study found similar infiltrations with immune cells in all groups, with the highest intensity in ARTMAL and lowest in UNS. Paolini et al. [46] also observed a higher number of inflammatory cells in the fundic part of the abomasa of goats infected with *H. contortus* that received tannins representing 5% of the dietary dry matter compared to a control group. The inflammatory infiltrates in our experiment were mainly localised in both the superficial and basal portions of the mucosa in all treated groups but only in the basal portions of the abomasal glands in UNS. Pérez et al. [47] found infiltrates of eosinophils and mononuclear cells mostly in the deep areas of the mucosa in goats infected with *H. contortus*, with fewer infiltrates in the submucosa. The expansion of immune cells to all layers of the mucosa in our study was probably affected by the stronger inflammation in these groups or by the immunomodulatory effect of the plant supplements. Five of the six animals in MAL were characterised by the presence of lymphoid nodules in the mucosa. Venturina et al. [48] frequently detected these lymphoid aggregates in susceptible lambs infected with the abomasal nematode *Teladorsagia circumcincta*. Salman and Duncan [49] also observed lymphoid aggregates at the base of the mucosa in sheep vaccinated against *H. contortus*, which was due to repeated immunisation and an improved immune response. In the present study, the nutrient supplementation with plants and their bioactive compounds probably affected local immune response in the abomasum.

The gastrointestinal tract is covered with a protective layer of mucus containing mainly mucins and other substances such as immunoglobulins, which can inhibit the growth of parasites or eliminate them from the tissue. This mucosal layer represents the first line of innate defense against GIN infections [50]. Increased secretion of mucus and the occurrence of inhibitory substances in the mucus have been associated with the development of immunity against GINs [51]. Pérez et al. [47] described an increase in mucus secretion in the abomasum together with the infiltration of immune cells into the abomasal mucosa of goats infected with *H. contortus*. Mucus in our experiment was produced most strongly throughout the thickness of the mucosa in ARTMAL, perhaps due to the bioactive compounds of the plants and the strongest intensity of inflammation in this group. ARTMAL also had the lowest worm count in the abomasa [19], which could have been caused by the elimination of parasites by the mucus.

The infiltration of abomasal tissue by eosinophils, mast cells, and other immune cells is a natural response to *H. contortus* infection in small ruminants. Eosinophilia is often associated with resistance [52] because eosinophils play an essential role in the host response against GINs by killing the parasites. The number of tissue eosinophils did not differ significantly between the experimental groups, perhaps because medicinal plants do not affect counts of tissue eosinophils. Paolini et al. [46] reported similar results, where dietary supplementation with condensed tannins did not affect the numbers of eosinophils in the abomasa of goats infected with *H. contortus*. The lack of significant differences between the experimental groups may also have been due to an inappropriately timed necropsy. Terefe et al. [53] reported no significant differences in the number of tissue eosinophils between susceptible and resistant breeds of lambs infected with *H. contortus*. They also suggested an inappropriate timing of necropsy, which was supported by the results of a haematological examination, where blood eosinophilia peaked between 10 and 20 d after infection. We did not analyse blood parameters, so we could not confirm this finding. Our previous studies, however, did not identify any changes in counts of blood eosinophils in groups treated with a mixture of medicinal plants [20,21], so we did not expect medicinal plants to positively affect counts of blood and tissue eosinophils.

Mast cells are involved in a variety of physiological functions, including parasite elimination [54]. Their expansion and distribution in tissues can change in response to helminth parasites [55]. The numbers of mucosal mast cells in our experiment were higher in ART and ARTMAL than MAL. Medicinal plants are a source of bioactive compounds, which can influence the natural immunity of an animal. For example, Tzamaloukas et al. [56] found that the local immune response against *T. circumcincta* improved in lambs that grazed on chicory and sulla. These lambs had higher numbers of mast cells and globular leucocytes in abomasal mucosa and a poorer development of nematodes. In our study, the lowest number of mast cells in the group treated with *M. sylvestris* may have been due to the anti-inflammatory properties of the bioactive compounds in this plant [40]. These anti-inflammatory properties may help to account for UNS having a higher number of mast cells than MAL. *H. contortus* infections are associated with an increase in plasma cells in abomasal tissue [57], and the presence of these cells has been linked to the production of nematode-specific antibodies, which play a role in protective immune responses [28]. In our experiment MAL had fewer plasma cells than did ARTMAL. Again, the anti-inflammatory effect of the *M. sylvestris* bioactive compounds may have affected the reduction of plasma cells in the abomasal tissue. Interestingly, the levels of IgG and IgA in the serum did not differ significantly between MAL and ARTMAL [19], but we assume that the difference in antibody production could be local, in the abomasal mucus. Analysing local antibody responses may thus be interesting.

Abomasal mucosa is often damaged during *H. contortus* infections, which can be due to the presence of the parasite or of reactive oxygen species, which arise during these infections. Tissues can regenerate under improved health conditions, indicated by an increase in mitotic figures in infected tissue. The number of mitotic figures in the abomasal mucosa in our experiment was higher in both MAL and ARTMAL than it was for UNS and ART. Both *A. absinthium* and *M. sylvestris* have strong antioxidant properties, and they may protect abomasal mucosa and promote mucosal healing by the elimination of free radicals. The thicker mucus in ARTMAL may also have reduced mucosal damage caused by reactive oxygen species because mucus can also act as an antioxidant [58]. *M. sylvestris* alone or in combination with *A. absinthium* may therefore enhance the regeneration of damaged tissue by their antioxidant properties. Dietary supplementation with these two medicinal plants may consequently influence the local immune response in the abomasum and affect the local status of antioxidants of an infected animal, but more research is needed.

## 5. Conclusions

This study represents the first report of the effects of medicinal plants (*A. absinthium* and *M. sylvestris*) on antioxidant parameters and local immune responses of abomasal mucosa of lambs infected with parasitic GINs. Treatment with the medicinal plants influenced antioxidant parameters in the abomasal mucosa and helped trigger a local immune response in tissues. We can conclude that dietary supplementation with medicinal plants may increase the resistance of lambs to infection with *H. contortus*. Further research, however, is needed to understand these effects of medicinal plants.

## Figures and Tables

**Figure 1 animals-11-00462-f001:**
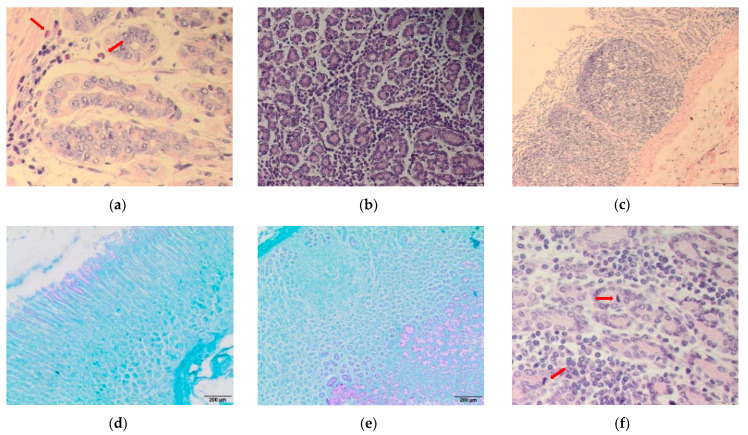
Histopathological sections of sheep abomasa infected with *Haemonchus contortus*: (**a**) section stained by haematoxylin and eosin (HE) (400×) showing infiltration with eosinophils (red arrows) in unsupplemented lambs (UNS); (**b**) section stained by HE (200×) showing strong inflammation in the mucosal membranes in lambs supplemented with a mixture of *A. absinthium* stems and *M. sylvestris* flowers (ARTMAL); (**c**) section stained by HE (100×) showing formation of lymphoid nodules in the mucous membrane in lambs supplemented with *M. sylvestris* flowers (MAL); (**d**) section stained by Periodic acid-Shiff (PAS) (40×) showing mucus formation (pink color) in the superficial part of the mucosal membrane in UNS; (**e**) section stained by PAS (40×) showing mucus formation (pink color) in all layers of the mucosal membrane and distention of superficial parts of glands in lambs supplemented with *A. absinthium* stems (ART); and (**f**) section stained by HE (200×) showing moderate mitotic activity of the epithelial cells (red arrows) and inflammatory infiltrate with mononuclear cells in ART.

**Figure 2 animals-11-00462-f002:**
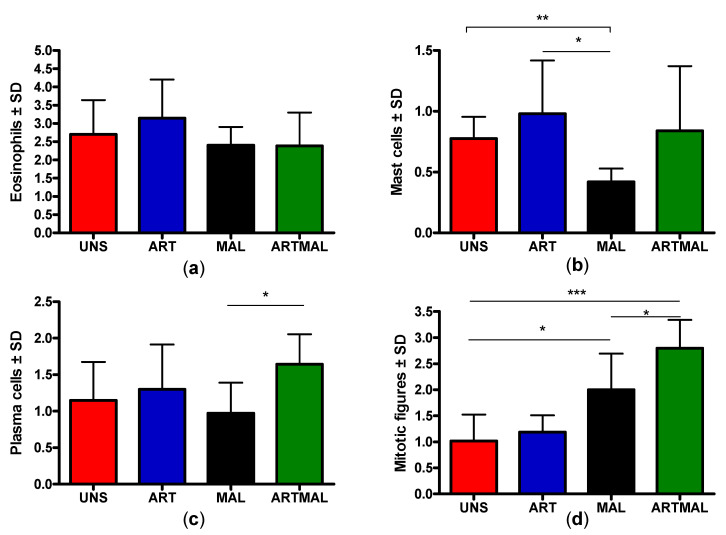
Mean numbers of different immune cells and mitotic figures in the abomasa of lambs infected with *Haemonchus contortus* fed with different diets (* *p* < 0.05; ** *p* < 0.01; *** *p* < 0.001): (**a**) mean numbers of eosinophils; (**b**) mean numbers of mast cells; (**c**) mean numbers of plasma cells; and (**d**) mean numbers of mitotic figures. UNS: unsupplemented lambs; ART: lambs supplemented with *A. absinthium* stems; MAL: lambs supplemented with *M. sylvestris* flowers; ARTMAL: lambs supplemented with a mixture of *A. absinthium* stems and *M. sylvestris* flowers.

**Table 1 animals-11-00462-t001:** Composition and concentration of the fractions of plant methanolic extracts.

Fraction (F)	RT (min)	Composition of Fraction	FW (mg/g)	Concentration (µg/mL)
*A. absinthium*				
F1	1.00–5.25	Very polar substances, sugars, polar peptides and proteins, polar phenolic acids	45.913	46,000
F2	5.25–18.00	Polyphenols, less polar phenolic acids, sesquiterpenes	19.813	20,000
F3	18.00–25.00	A similar spectrum of substances as F2, but of lower polarity	5.583	6000
F4	25.00–45.00	Nonpolar substances, lipids, resins, di- and triterpenes	12.102	12,000
*M. sylvestris*				
F1	1.00–5.25	Sugars, polar phenols	150.441	150,000
F2	5.25–10.50	Anthocyanins, other phenolic compounds	41.818	42,000
F3	10.50–17.60	Polyphenols, flavonoids, phenolic acids	9.403	9000
F4	17.60–45.00	Nonpolar substances, esters, fatty acids, lipids	18.368	18,000

RT: retention time of collection; FW: fresh weight.

**Table 2 animals-11-00462-t002:** Antioxidant parameters in the abomasal mucosa and sera of infected lambs.

Item	Treatment					
	UNS	ART	MAL	ARTMAL	SEM	*p-*Value
**Abomasum**						
MDA (nmol/g protein)	277.71	237.69	209.35	174.78 *	13.67	0.048
GPx (U/g protein)	98.46	88.48	92.63	95.67	2.314	0.508
TAC (µmol/g protein)	50.98	51.69	62.94 *	55.65	2.381	0.050
**Serum**						
MDA (μmol/L)	0.188	0.192	0.185	0.160	0.008	0.498
GPx (U/mL)	0.106	0.095	0.102	0.102	0.007	0.957
TAC (mmol/L)	0.593	0.512	0.578	0.542	0.013	0.102

MDA: malondialdehyde; GPx: glutathione peroxidase activity; TAC: total antioxidant capacity; UNS: unsupplemented lambs; ART: lambs supplemented with *A. absinthium* stems; MAL: lambs supplemented with *M. sylvestris* flowers; ARTMAL: lambs supplemented with a mixture of *A. absinthium stems* and *M. sylvestris* flowers; SD: standard deviation. * *p* < 0.05 compared to UNS.

## Data Availability

Data will be made available, upon reasonable request to the corresponding author.

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
