# Peer review of "Effect of Artemisia absinthium and Malva sylvestris on Antioxidant Parameters and Abomasal Histopathology in Lambs Experimentally Infected with Haemonchus contortus"

_animals, 2021, doi:10.3390/ani11020462_

Round 1

Reviewer 1 Report

This paper evaluated the effect of medical plants on antioxidant parameters and abomasal histopathology in lambs infected with Haemonchus contortus. It is meaningful for the development of antibiotic free methods to deal with the GIN. The manuscript language is good. A few of comments as follows:

Line 111: how long was the adaptation period lasted?

line 124-127: What was the ratio to mix the ART and MAL in ARTMAL group?It doesn’t look like 1g ART plus 15g MAL for ARTMAL group according to the phytochemical content.

Table 2: show us the SEM instead of SD.

Author Response

Dear Reviewer,          
attached you can find the revised Manuscript ID: animals-1060547.

Thank you very much for the valuable comments and suggestions. Detailed responses to each point were clearly highlighted by red color in the revised manuscript. We have tried to address your all comments by carefully revising the manuscript in light of your comments. We hope our corrections and additions have improved the manuscript sufficiently for publication.

Dominika  Mravčáková, behalf all of the co-authors

R1: How long was the adaptation period lasted?

AUTHORS: Lines 105-108: The adaptation period lasted for 14 days. The sentence was changed as follows: ”Briefly, twenty-four 3–4-month-old female lambs (Improved Valachian) with initial body weights of 18.67 ± 0.55 kg were housed in common stalls on a commercial sheep farm (Hodkovce, Slovakia) for 14 d for acclimatization to the feeding and maintained in their production system during the experiment with free access to water.”

R1: What was the ratio to mix the ART and MAL in ARTMAL group? It doesn’t look like 1g ART plus 15g MAL for ARTMAL group according to the phytochemical content.

AUTHORS:

The dry plant samples (wormwood, mallow, and their mix) were ground to a fine powder and 100 mg were extracted three times with 80% MeOH for 30 min at 40 °C each. The phytochemical content analysis of dry plants (individual and mix) in detail was described in the study Mravčáková et al. (2020). The results of phenolic acids and flavonoids of the plant materials were analyzed by an ultra-high-resolution mass spectrometry (UHRMS) (Dionex UltiMate 3000RS, Thermo Scientific, Darmstadt, Germany) system with a high-resolution quadrupole time-of-flight mass spectrometer (HR/Q-TOF/MS, Compact, Bruker Daltonik GmbH, Bremen, Germany).

Lines 121-123: To avoid misinterpretation the sentence was changed as follows: “The phytochemical substances of A. absinthium contained 6.48 g/kg DM phenolic acids and 0.35 g/kg DM flavonoids, M. sylvestris contained 0.65 g/kg DM phenolic acids and 6.48 g/kg DM flavonoids.

Reference:

Mravčáková, D.; Komáromyová, M.; Babják, M.; Urda Dolinská, M.; Königová, A.; Petrič, D.; Čobanová, K.; Śluzarcyk, S.; Cieslak, A.; Várady, M.; Váradyová, Z. Anthelmintic activity of wormwood (Artemisia absinthium L.) and mallow (Malva sylvestris L.) against Haemonchus contortus in sheep. Animals. 2020, 10, 1–15.

R1: Table 2: Show us the SEM instead of SD.

AUTHORS: We changed the SD values into the SEM values in Table 2.

Reviewer 2 Report

The study of medicinal plants has become very popular in recent years as anthelmintic, that is why I find this work very interesting "Effect of Artemisia absinthium and Malva sylvestris on antioxidant parameters and abomasal histopathology in lambs experimentally infected with Haemonchus contortus" and I find it well structured (introduction) but I have some questions;

Correct the following, if the plant is spoken as the common name, place the scientific name followed in parentheses for example. Wormwood (Artemisia absinthium).

It is confused if I do aqueous or methanolic extract

Mention the route of administration, how the extracts were administered

There are NO positive and negative controls to compare

Why fraction and not evaluate the fractions? Or use an in vitro model

They could discuss secondary metabolites like flavonoids, phenolic acids, etc., these articles can help you

Edgar Jesus Delgado-Nuñez, Alejandro Zamilpa, González-Cortazar Manasés, et al., Isorhamnetin: A nematocidal from Prosopis laevigata leaves against Haemonchus contortus eggs and larvae. Biomolecules 2020; 10 (5): 773.

Agustín Olmedo-Juárez, Miguel Angel Zarza-Albarran, Rolando Rojo-Rubio, et al., Acacia farnesiana pods (plant: Fabaceae) possesses anti-parasitic compounds against Haemonchus contortus in female lambs. Experimental Parasitology, 2020; 218; 107980.

Castillo-Mitre GF, Olmedo-Juárez A, Rojo-Rubio R, et al., Caffeoyl and coumaroyl derivatives from Acacia cochliacantha exhibit ovicidal activity against Haemonchus contortus. Journal Ethnopharmacology. 204; 125-131: 2017.

Attach the HPLC chromatograms as supplemental data

Author Response

Dear Reviewer,          
attached you can find the revised Manuscript ID: animals-1060547.

Thank you very much for the valuable comments and suggestions. Detailed responses to each point were clearly highlighted by red color in the revised manuscript. We have tried to address your all comments by carefully revising the manuscript in light of your comments. We hope our corrections and additions have improved the manuscript sufficiently for publication.

Dominika  Mravčáková, behalf all of the co-authors

R2: Correct the following, if the plant is spoken as the common name, place the scientific name followed in parentheses for example. Wormwood (Artemisia absinthium).

AUTHORS:

Lines 35-36: We added scientific name of plants in parentheses after their common name.

Line 93: We added scientific name of plants in parentheses after their common name.

R2: It is confused if I do aqueous or methanolic extract.

AUTHORS: Lines 111-115: We were not working with plants aqueous or methanolic extracts but with dry parts of medicinal plants, similarly as in our previous studies (Váradyová et al., 2017, 2018a; Mravčáková et al., 2019). In our present study, we chose these two plants base on our previous in vitro study (Váradyová et al., 2018b), in which the wider spectrum of medicinal plant extracts exhibited stronger ovicidal and larvicidal activity, but the most efficient were A. absinthium and M. sylvestris. However, our previous results concerning A. absinthium and M. sylvestris (Mravčáková et al., 2020) revealed that the strong anthelmintic effect of both these medicinal plants observed in vitro was not fully confirmed in vivo. This is because it probably depends on the variety and synergy of bioactive compounds of a combination of several (more than two in our case) medicinal plants that together affect and contributes to a certain pharmacological efficacy (as was observed in our previous studies with mixes of 9-13 medicinal plants, Váradyová et al., 2017, 2018a; Mravčáková et al., 2019)

References:

Váradyová, Z.; Kišidayová, S.; Čobanová, K.; Grešáková, Ľ.; Babják, M.; Königová, A.; Urda Dolinská, M.; Várady, M. The impact of a mixture of medicinal herbs on ruminal fermentation, parasitological status and hematological parameters of the lambs experimentally infected with Haemonchus contortus. Small Rumin Res. 2017, 151, 124–132.

Váradyová, Z.; Mravčáková, D.; Babják, M.; Bryszak, M.; Grešáková, Ľ.; Čobanová, K.; Kišidayová, S.; Plachá, I.; Königová, A.; Cieslak, A.; Ślusaczyk, S.; Pecio, L.; Kowalczyk, M.; Várady, M. Effects of herbal nutraceuticals and/or zinc against Haemonchus contortus in lambs experimentally infected. BMC Vet Res. 2018a, 14, 78.

Mravčáková, D.; Váradyová, Z.; Kopčáková, A.; Čobanová, K.; Grešáková, Ľ.; Kišidayová, S.; Babják, M.; Urda Dolinská, M.; Dvorožňáková, E.; Königová, A.; Vadlejch, J.; Cieslak, A.; Śluzarcyk, S.; Várady, M. Natural chemotherapeutic alternatives for controlling of haemonchosis in sheep. BMC Vet Res. 2019, 15, 302.

Váradyová, Z.; Pisarčíková, J.; Babják, M.; Hodges, A.; Mravčáková, D.; Kišidayová, S.; Königová, A.; Vadlejch, J.; Várady, M. Ovicidal and larvicidal activity of extracts from medicinal-plants against Haemonchus contortus. Experimental Parasitology 2018b, 195, 71–77)

R2: Mention the route of administration, how the extracts were administered.

AUTHORS: Lines 117-119: The dried plant components of A. absinthium and M. sylvestris both from commercial sources (AGROKARPATY, Plavnica, Slovak Republic) were mixed daily with the oats during the experimental period (75 d).

R2: There are NO positive and negative controls to compare

AUTHORS: Lines 111-115: The current study is only part of a larger study that investigated natural chemotherapeutic alternatives for controlling haemonchosis in lambs and all experiment had been described in more details in paper: Mravčáková et al., 2020: (“Anthelmintic activity of wormwood (Artemisia absinthium L.) and mallow (Malva sylvestris L.) against Haemonchus contortus in sheep”). The experiment was a predominantly parasitological study. From a parasitological point of view, a negative control (uninfected animals) is not necessary because (group) animals without infection are not comparable with infected groups. As a positive control, we used infected unsupplemented lambs, which did not obtain any medicinal plant treatment.

R2: Why fraction and not evaluate the fractions? Or use an in vitro model.

AUTHORS: We agree with the reviewer, but the current experiment is only part of a larger parasitological study and the bioactive compounds of the plant materials were previously in more detail analyzed by UHRMS and described in Mravčáková et al., (2020). The present study was concerning mainly to evaluate the effect of diets with dry wormwood and mallow mainly on antioxidant parameters and abomasal tissues histopathological changes in the lambs with H. contortus. The use of the fraction in vitro model was not the subject of the present study. The fraction analyses were for us just a pilot analysis for comment a list of the main groups of secondary metabolites in both plants.

Reference:

Mravčáková, D.; Komáromyová, M.; Babják, M.; Urda Dolinská, M.; Königová, A.; Petrič, D.; Čobanová, K.; Śluzarcyk, S.; Cieslak, A.; Várady, M.; Váradyová, Z. Anthelmintic activity of wormwood (Artemisia absinthium L.) and mallow (Malva sylvestris L.) against Haemonchus contortus in sheep. Animals. 2020, 10, 1–15.

R2: They could discuss secondary metabolites like flavonoids, phenolic acids, etc., these articles can help you:

Edgar Jesus Delgado-Nuñez, Alejandro Zamilpa, González-Cortazar Manasés, et al., Isorhamnetin: A nematocidal from Prosopis laevigata leaves against Haemonchus contortus eggs and larvae. Biomolecules 2020; 10 (5): 773.

Agustín Olmedo-Juárez, Miguel Angel Zarza-Albarran, Rolando Rojo-Rubio, et al., Acacia farnesiana pods (plant: Fabaceae) possesses anti-parasitic compounds against Haemonchus contortus in female lambs. Experimental Parasitology, 2020; 218; 107980.

Castillo-Mitre GF, Olmedo-Juárez A, Rojo-Rubio R, et al., Caffeoyl and coumaroyl derivatives from Acacia cochliacantha exhibit ovicidal activity against Haemonchus contortus. Journal Ethnopharmacology. 204; 125-131: 2017.

AUTHORS: Thank you for your advice. The authors agree with the reviewer that the results of secondary metabolites could discuss. The topic could be much more complex than what emerges from the discussion and conclusions of this study. However, the present study is concerning predominantly to evaluate the effect of diets containing dry wormwood and mallow on antioxidant parameters in sera and abomasal mucosa and on histopathological changes in the abomasal tissues of lambs experimentally infected with H. contortus.

Lines 66-68: We included in the Introduction: “Several studies are currently dealing with bioactive compounds with the anthelmintic activity of organic fractions obtained from plant extracts used as natural anthelmintics against H. contortus eggs and infective larvae [16-18]“ (Castillo-Mitre et al., 2017; Delgado-Nuñez et al., 2020; Olmedo-Juárez et al., 2020).

Lines 256-261: We included in the Discussion: “More complex studies are probably necessary before excluding the direct anthelminthic properties of these plants in vivo [19] because the complexity of the physiology of the gastrointestinal system of ruminants makes it particularly difficult to evaluate the potential efficacy of the same plant when administered fresh after the harvest or after its drying. In addition, promising results are also shown by exploring the potential anthelmintic effect of fractions of bioactive compounds from extracts of some plants (e.g., Acacia cochliacantha, A. farnesiana, Prosopis laevigata) in vitro or in vivo [16-18].

Lines 418-427: We included references for discussed research:

Castillo-Mitre, G.F; Olmedo-Juárez, A.; Rojo-Rubio, R.; Gonzáles-Cortázar, M.; Mendoza-de Gives, P.; Hernández-Beteta. E.E.; Reyes-Guerrero, D.E.; López-Arellano, M.E.; Vázquez-Armijo, J.F.; Ramírez-Vargas, G.; Zamilpa, A. Caffeoyl and coumaroyl derivates from Acacia cochliacantha exhibit ovicidal activity against Haemonchus contortus. J. Ethnopharmacol. 2017, 204, 125-131. doi:10.1016/j.jep.2017.04.010

Delgado-Núñez, E.J.; Zamilpa, A.; Gonzáles-Cortazar, M.; Olmedo-Juárez. A.; Cardoso-Taketa, A.; Sánchez-Mendoza, E.; Tapia-Maruri, D.; Salinas-Sánchez, D.O.; Mendoza-de Gives, P. Isorhamnetin: A nematocidal flavonoid from Prosopis Leavigata leaves against Haemonchus contortus eggs and larvae. Biomolecules. 2020, 10, 733. doi: 10.3390/biom10050773

Olmedo-Juárez, A.; Zarza-Albarran, M.A.; Rojo-Rubio, R.; Zamilpa, A.; Gonzáles-Cortazar, M.; Mondragón-Ancelmo, J.; Rivero-Péerez, N.; Mendoza-de Gives, P. Acacia farnesiana pods (plant: Fabaceae) possesses anti-parasitic compounds against Haemonchus contortus in female lambs. Exp Parasitol. 2020, 218, 107980. doi:10.1016/j.exppara.2020.107980

R2: Attach the HPLC chromatograms as supplemental data.

AUTHORS: We attached the HPLC chromatograms as supplemental data.

Reviewer 3 Report

The study “Effect of Artemisia absinthium and Malva sylvestris on antioxidant parameters and abomasal histopathology in lambs experimentally infected with Haemonchus contortus” represents, according to the authors “The first report of the effects of medicinal plants (A. absinthium and M. sylvestris) on antioxidant parameters and local immune responses of abomasal mucosa of lambs infected with parasitic GINs. Treatment with the medicinal plants influenced antioxidant parameters in the abomasal mucosa and helped trigger a local immune response in tissues”. The authors concluded “that dietary supplementation with medicinal plants may increase the resistance of lambs to infection with Haemonchus contortus”.

The manuscript is well written, well-structured and balanced. My opinion is that the manuscript has quality to be published in Animals but there are some minor aspects that I would like to clarify/suggest with the authors before its publication. 

Comment 1: I suggest presenting a list of abbreviations at the beginning of the manuscript.

ABSTRACT

Comment 2 (lines 50-51):  As it has not yet been said what these acronyms mean, authors should explain ARTMAL, MAL, UNS and ART.

Comment 3 (lines 56-57): I think these would be 3 strong keywords: Artemisia absinthium, Malva sylvestris, Haemonchus contortus

INTRODUCTION

Comment 4: It is just a suggestion, but I think it would be interesting to say anything about the applicability in real situations. I ask: would be the use of these plants in a natural situation of grazing (are they palatable and consumed/chosen by the animals?)? Or after harvest, possibly making this use unfeasible from an economic point of view? (Only for this reason I selected Significance of Content “Average”, but, obviously “High” from a scientific point of view).        

MATERIAL AND METHODS

Comment 5: As the authors referThis experiment was part of a study investigating the anthelmintic activity of A. absinthium and 257 M. sylvestris against H. contortus in lambs [Mravčáková et al. Animals. 2020; doi:10.3390/ani10020219]”. On the other hand, and in relation to other methodologies, they refer to another work Váradyová et al. BMC Vet Res. 2018; doi:10.1186/s12917-018-1405-4.  I see no problem with that, but I am of the opinion, and in order not to force the reader to consult all works, provide a minimum of information. For example, and regarding blood collection (lines 127-131), the authors should mention the time of the day and how the samples were obtained.

Comment 6 (lines 116-117): Why were used these levels (1 g DM/d/lamb and 15 g DM/d/lamb)? Were determined in previous works? Do they have to do with palatability and/or toxicity issues?

Comment 7 (131-134): I can understand why these values appear here, but honestly I think they are results and should appear in that section, perhaps at the beginning, or else, associated with one of the sub-chapters of the determined parameters. “Mean worm abomasal counts were 899 ± 95, 984 133 ± 382, 876 ± 269 and 787 ± 335 for UNS, ART, MAL and ARTMAL, respectively [31]”.

RESULTS

Comment 8: Authors should try to improve the quality of photomicrographs

Comment 9 (Fig. 1): The captions of the photomicrographs should always be aware that some readers may not dominate the histology and, in this sense, they must have the necessary notations that identify/locate the findings to which they refer.

Author Response

Dear Reviewer,          
attached you can find the revised Manuscript ID: animals-1060547.

Thank you very much for the valuable comments and suggestions. Detailed responses to each point were clearly highlighted by red color in the revised manuscript. We have tried to address your all comments by carefully revising the manuscript in light of your comments. We hope our corrections and additions have improved the manuscript sufficiently for publication.

Dominika  Mravčáková, behalf all of the co-authors

R3: I suggest presenting a list of abbreviations at the beginning of the manuscript.

AUTHORS: A list of abbreviations (independently) is not standard on the first page according to Instructions for Authors of Animals. However, we defined missing abbreviations at their first mention there, where they were missing in the Abstract and elsewhere.

R3: Abstract:  

Lines 50-51: As it has not yet been said what these acronyms mean, authors should explain ARTMAL, MAL, UNS and ART.

AUTHORS: Line 39-41: We added acronyms UNS, ART, MAL and ARTMAL into the abstract for each experimental group.

R3: Lines 56-57: I think these would be 3 strong keywords: Artemisia absinthium, Malva sylvestris, Haemonchus contortus.

AUTHORS: Lines 50-51: The previous keywords:abomasum; antioxidant parameters; gastrointestinal nematode parasite; histopathological changes; local immune response; sheep”

we changed according to R3 comments:

“abomasum; antioxidant parameters; Artemisia absinthium; gastrointestinal nematode parasite Haemonchus contortus; histopathological changes; local immune response; Malva sylvestris.”

R3: Introduction: It is just a suggestion, but I think it would be interesting to say anything about the applicability in real situations. I ask: would be the use of these plants in a natural situation of grazing (are they palatable and consumed/chosen by the animals?)? Or after harvest, possibly making this use unfeasible from an economic point of view? (Only for this reason I selected Significance of Content “Average”, but, obviously “High” from a scientific point of view).        

AUTHORS: The current study is only part of a larger study that investigated natural chemotherapeutic alternatives for controlling haemonchosis in lambs and all experiment had been described in more details in the paper: Mravčáková et al., 2020: (“Anthelmintic activity of wormwood (Artemisia absinthium L.) and mallow (Malva sylvestris L.) against Haemonchus contortus in sheep”). The all experiment was a predominantly parasitological study in which we hypothesized that some medicinal plants are by themselves multicomponent mixes and could elicit effects via pharmacological activity on H. contortus infected lambs. The medicinal plants wormwood and mallow were chosen based on their previously described best phytotherapeutic properties and anthelmintic activity in vitro (Váradyová et al., 2018a). The conclusion from our study (Mravčáková et al., 2020) is also “about the applicability of our results in real situations” and showed that using medicinal plants, even those with the best anthelmintic properties in vitro, may not have sufficient effects in vivo on H. contortus infected lamb.

Lines 66-74: We included in the Introduction: “Several studies are currently dealing with bioactive compounds with the anthelmintic activity of organic fractions obtained from plant extracts used as natural anthelmintics against H. contortus eggs and infective larvae [16-18] (Castillo-Mitre et al., 2017; Delgado-Nuñez et al., 2020; Olmedo-Juárez et al., 2020). However, the use of whole plants as dietary supplements in feeds can lead to different results. Our recent study [19] (Mravčáková et al., 2020) showed that the A. absinthium and M. sylvestris possess a strong anthelmintic effect in vitro versus a slight effect in vivo. On the other hand, the variety and synergy of bioactive compounds of several medicinal plants and their different combinations together described in our previous studies can contribute to a certain pharmacological efficacy [20-22]. (Váradyová et al., 2017, 2018b; Mravčáková et al., 2019)”. These studies concluded that supplementation with a mixture of dried medicinal plants in the diet of lambs could slow the dynamics of infection and increase the resistance of lambs to parasitic infection.”

References:

Mravčáková, D.; Komáromyová, M.; Babják, M.; Urda Dolinská, M.; Königová, A.; Petrič, D.; Čobanová, K.; Śluzarcyk, S.; Cieslak, A.; Várady, M.; Váradyová, Z. Anthelmintic activity of wormwood (Artemisia absinthium L.) and mallow (Malva sylvestris L.) against Haemonchus contortus in sheep. Animals. 2020, 10, 1–15.

Váradyová, Z.; Pisarčíková, J.; Babják, M.; Hodges, A.; Mravčáková, D.; Kišidayová, S.; Königová, A.; Vadlejch, J.; Várady, M. Ovicidal and larvicidal activity of extracts from medicinal-plants against Haemonchus contortus. Experimental Parasitology 2018a, 195, 71–77)

Váradyová, Z.; Kišidayová, S.; Čobanová, K.; Grešáková, Ľ.; Babják, M.; Königová, A.; Urda Dolinská, M.; Várady, M. The impact of a mixture of medicinal herbs on ruminal fermentation, parasitological status and hematological parameters of the lambs experimentally infected with Haemonchus contortus. Small Rumin Res. 2017, 151, 124–132.

Váradyová, Z.; Mravčáková, D.; Babják, M.; Bryszak, M.; Grešáková, Ľ.; Čobanová, K.; Kišidayová, S.; Plachá, I.; Königová, A.; Cieslak, A.; Ślusaczyk, S.; Pecio, L.; Kowalczyk, M.; Várady, M. Effects of herbal nutraceuticals and/or zinc against Haemonchus contortus in lambs experimentally infected. BMC Vet Res. 2018b, 14, 78.

Mravčáková, D.; Váradyová, Z.; Kopčáková, A.; Čobanová, K.; Grešáková, Ľ.; Kišidayová, S.; Babják, M.; Urda Dolinská, M.; Dvorožňáková, E.; Königová, A.; Vadlejch, J.; Cieslak, A.; Śluzarcyk, S.; Várady, M. Natural chemotherapeutic alternatives for controlling of haemonchosis in sheep. BMC Vet Res. 2019, 15, 302.

Váradyová, Z.; Pisarčíková, J.; Babják, M.; Hodges, A.; Mravčáková, D.; Kišidayová, S.; Königová, A.; Vadlejch, J.; Várady, M. Ovicidal and larvicidal activity of extracts from medicinal-plants against Haemonchus contortus. Experimental Parasitology 2018b, 195, 71–77)

R3: Material and methods:

Lines 127-131: As the authors refer “This experiment was part of a study investigating the anthelmintic activity of A. absinthium and M. sylvestris against H. contortus in lambs [Mravčáková et al. Animals. 2020; doi:10.3390/ani10020219]”. On the other hand, and concerning other methodologies, they refer to another work Váradyová et al. BMC Vet Res. 2018; doi:10.1186/s12917-018-1405-4.  I see no problem with that, but I think, and in order not to force the reader to consult all works, provide a minimum of information. For example, and regarding blood collection (lines 127-131), the authors should mention the time of the day and how the samples were obtained.

AUTHORS: Lines 123-126: We added into the section Material and methods, subsection Animals and Experimental Design: “Blood samples for sera were collected from each animal at day 75 from the jugular vein by using a 21-gauge needle and syringe into 10 mL serum separator tubes (Sarstedt AG & Co, Nümbrecht, Germany) and centrifuged at 1200 g for 10 min at room temperature. The sera were stored at –80 °C until analysis.”

R3: Material and methods: Lines 116-117: Why were used these levels (1 g DM/d/lamb and 15 g DM/d/lamb)? Were determined in previous works? Do they have to do with palatability and/or toxicity issues?

AUTHORS: The amount of used medicinal plants was based on our previous studies (see reference Mravčáková et al., 2020) The feeding trial lasted 75 days preceded by a 14-day adaptation period to the pens, palatability of diets and experimental conditions. The dried plant components of A. absinthium and M. sylvestris were mixed daily with the oats during the experimental period The effect of individual medicinal plants and mixes (including A. absinthium and M. sylvestris on rumen fermentation parameters and rumen protozoan population were done previously (Váradyová et al., 2017, Petrič et al., 2020).

References:

Petrič, D., Mravčáková, D., Kucková, K., Čobanová, K., Kišidayová, S., Cieslak, A., Slusarczyk, S., Váradyová, Z. Effect of dry medicinal plants (wormwood, chamomile, fumitory and mallow) on in vitro ruminal antioxidant capacity and fermentation patterns of sheep. In Journal of Animal Physiology and Animal Nutrition. 2020, vol. 104, p. 1219–1232 

Váradyová, Z.; Kišidayová, S.; Čobanová, K.; Grešáková, Ľ.; Babják, M.; Königová, A.; Urda Dolinská, M.; Várady, M. The impact of a mixture of medicinal herbs on ruminal fermentation, parasitological status and hematological parameters of the lambs experimentally infected with Haemonchus contortus. Small Rumin Res. 2017, 151, 124–132.

R3: Material and methods: Lines 131-134: I can understand why these values appear here, but honestly I think they are results and should appear in that section, perhaps at the beginning, or else, associated with one of the sub-chapters of the determined parameters. “Mean worm abomasal counts were 899 ± 95, 984 133 ± 382, 876 ± 269 and 787 ± 335 for UNS, ART, MAL and ARTMAL, respectively [31]”.

AUTHORS: Lines 131-134: We removed the sentence: “Mean worm abomasal counts were 899 ± 95, 984 133 ± 382, 876 ± 269 and 787 ± 335 for UNS, ART, MAL and ARTMAL, respectively [31]” from Material and Methods section, subsection Animals and Experimental Design.

Lines 207-208: We added this sentence in the Results section, in the subsection Abomasum Histology.

R3: Results: Authors should try to improve the quality of photomicrographs.

AUTHORS: According to Instructions for Authors of Animals all Figures should be inserted into the main text close to their first citation. But I will try to attach the Figures to the Journal also separately to improve the quality of photomicrographs if it will be possible.

R3: Results: Fig. 1: The captions of the photomicrographs should always be aware that some readers may not dominate the histology and, in this sense, they must have the necessary notations that identify/locate the findings to which they refer.

AUTHORS: Fig. 1:  We added red arrows and short notations to describe findings in photomicrographs.

Round 2

Reviewer 2 Report

Interesting work and appropriate to be published
Answered all the questions
I propose for the future to do the phytochemical study with an in vitro model